# Estrogen Mediates the Sexual Dimorphism of GT1b-Induced Central Pain Sensitization

**DOI:** 10.3390/cells12050808

**Published:** 2023-03-06

**Authors:** Jaesung Lee, Seohyun Chung, Minkyu Hwang, Yeongkag Kwon, Seung Hyun Han, Sung Joong Lee

**Affiliations:** 1Department of Neuroscience and Physiology, Dental Research Institute, School of Dentistry, Seoul National University, Seoul 08826, Republic of Korea; 2Interdisciplinary Program in Neuroscience, College of Natural Science, Seoul National University, Seoul 08826, Republic of Korea; 3Department of Brain and Cognitive Sciences, College of Natural Sciences, Seoul National University, Seoul 08826, Republic of Korea; 4Department of Oral microbiology and Immunology, Dental Research Institute, School of Dentistry, Seoul National University, Seoul 08826, Republic of Korea; 5Research Division for Radiation Science, Korea Atomic Energy Research Institute, Jeongeup 56212, Republic of Korea

**Keywords:** GT1b, pain central sensitization, estrogen, sexual dimorphism, IL-1β

## Abstract

We have previously reported that the intrathecal (i.t.) administration of GT1b, a ganglioside, induces spinal cord microglia activation and central pain sensitization as an endogenous agonist of Toll-like receptor 2 on microglia. In this study, we investigated the sexual dimorphism of GT1b-induced central pain sensitization and the underlying mechanisms. GT1b administration induced central pain sensitization only in male but not in female mice. Spinal tissue transcriptomic comparison between male and female mice after GT1b injection suggested the putative involvement of estrogen (E2)-mediated signaling in the sexual dimorphism of GT1b-induced pain sensitization. Upon ovariectomy-reducing systemic E2, female mice became susceptible to GT1b-induced central pain sensitization, which was completely reversed by systemic E2 supplementation. Meanwhile, orchiectomy of male mice did not affect pain sensitization. As an underlying mechanism, we present evidence that E2 inhibits GT1b-induced inflammasome activation and subsequent IL-1β production. Our findings demonstrate that E2 is responsible for sexual dimorphism in GT1b-induced central pain sensitization.

## 1. Introduction

Neuropathic pain is a chronic pathological pain caused by damage or dysfunction in the nervous system [1]. The clinical symptoms of neuropathic pain include spontaneous pain, allodynia, and hyperalgesia [2], but its severity and prevalence vary depending on sex [3]. Increasing evidence based on animal studies shows that neuroimmune mechanisms underlie the sexual dimorphism of neuropathic pain [3,4]. For example, spinal cord microglia activation is required for central pain sensitization after peripheral nerve injury in male rodents [5]. However, it is dispensable for the induction of neuropathic pain in female rodents. Instead, adaptive-immune-cell infiltration, most likely T lymphocytes, into the spinal cord contributes to the development of nerve-injury-induced neuropathic pain in female rodents [6]. In addition, intrathecal (i.t) administration of LPS or CSF-1 induces spinal cord microglia activation and subsequent pain sensitization only in male mice, although it induced comparable levels of morphological microglia activation in the female spinal cord [7,8,9]. Therefore, spinal cord microglia play a distinct role in the development of central pain sensitization depending on sex; the activation of only male microglia, not female, renders central pain sensitization. Thus, it is not clear why female spinal cord microglia activation is unable to induce central pain sensitization. Studies suggest that spinal cord microglia display a distinct sex-specific molecular signature after peripheral nerve injury [10], and regulatory T-cells infiltrating into the spinal cord after nerve injury render differential microglia activation in female mice [8]. In another study, testosterone, a male sex hormone, was implicated in the pain-inducing signature of activated microglia in male rodents [6]. Therefore, the exact mechanisms of sexual dimorphism in spinal cord microglia activation and pain sensitization remain to be elucidated.

GT1b is one of the four major gangliosides of the CNS that constitute neuronal membrane lipid raft. We previously reported that GT1b is upregulated in damaged sensory neurons and transported to the spinal cord dorsal horn [11], which is required for the nerve-injury-induced spinal cord microglia activation and central pain sensitization [12,13]. Likewise, i.t. GT1b administration induced central pain sensitization [11]. Yet, the pain-inducing effect of i.t. GT1b administration was observed only in male mice [9]. In this study, we investigated the sexual dimorphism of GT1b-induced pain sensitization and the underlying mechanisms. Our data reveal that estrogen, a female sex hormone, is responsible for the sexual dimorphism of GT1b-induced spinal cord microglia activation and subsequent central pain sensitization.

## 2. Materials and Methods

### 2.1. Animals

The animal experiments were approved by the Institutional Animal Care and Use Committee (IACUC) of Seoul National University. Male and female C57BL/6J mice (8~10 weeks of age) were purchased from Daehan Biolink (DBL, Eumsung, Korea), and all animals were housed and maintained in a controlled environment at 22–24 °C and 55% humidity with a 12 h light/dark cycle in a specific pathogen-free (SPF) environment. They were provided with access to food and water ad libitum. All the protocols were performed in accordance with the guidelines from the International Association for the Study of Pain.

### 2.2. Intrathecal Injection

The mice were anesthetized with isoflurane in an O_2_ carrier (induction at 2% and maintenance at 1.5%), and the GT1b (25 μg/5 μL; Matreya LLC, Cat # 1548, State College, PA, USA) in saline solution was administered using a 10-μL Hamilton syringe (Hamilton Company, Cat # 701LT, Reno, NV, USA) with a 30 g needle as previously described [14]. The success of intrathecal injection was assessed by monitoring a slight tail-flick when the needle penetrated the subarachnoid space.

### 2.3. Immunohistochemistry

The mice were transcardially perfused with 0.1 M phosphate buffer (pH 7.4) followed by 4% paraformaldehyde, and the L5 spinal cord was removed and post-fixed in the same solution at 4 °C overnight. The spinal cord samples were transferred to 30% sucrose for at least 48 h and coronally cut into 16-μm-thick sections using a cryostat (Leica, Cat # CM1860, Wetzlar, Germany). The spinal cord sections were blocked in a solution containing 5% normal goat serum, 2.5% bovine serum albumin (BSA), and 0.2% Triton X-100 for 1.5 h at room temperature. Then, the spinal cord sections were incubated with rabbit anti-Iba-1 antibody (1:1000; Wako, Cat # 019-19741, Osaka, Japan). After rinsing 5 times with 0.1 M PBS, the samples were incubated with CY3-conjugated secondary antibodies (1:200; Jackson ImmunoResearch Laboratories, Cat # 111-165-003, West Grove, PA, USA) for 1.5 h at room temperature. The samples were mounted on glass slides with a Vectashield mounting medium (Vector Laboratories, Cat # H-1000-10, Burlingame, CA, USA).

### 2.4. Confocal Microscopy and 3D IMARIS Analysis

The images were captured using an LSM 800 confocal microscope (Carl Zeiss, Oberkochen, Germany). For the 3D reconstruction of the microglia, we took Z-stack images (6 μm depth, 460 μm steps) of the spinal dorsal horns using an LSM 800 (1024 × 1024 pixels, 16-bit depth, 0.624 mm pixel size). The raw image files (.czi) were converted and analyzed using IMARIS (Version 9.8.0, Oxford Instruments, Abingdon, UK). The morphology of the single microglia (from 4 to 6 mice/group) was analyzed using the Filament Tracer Tool with the following settings: Autopath algorithm; Dendrite starting point diameter, 16.3 μm; and Dendrite seed point diameter, 1 μm.

### 2.5. Behavioral Tests

The mechanical allodynia tests were performed as previously reported [11]. All the behavior tests occurred between 10:00 a.m. and 3:00 p.m., and the experimenter was blind to group assignments throughout the experiment. The mechanical sensitivity of the right hind paw was assessed using a calibrated series of von Frey hairs (0.02–6 g; Stoelting, Wood Dale, IL, USA), following an up-down method [15]. The thermal sensitivity was determined by measuring the paw withdrawal latencies in response to radiant heat [16]. Rapid paw withdrawal, licking, and flinching were interpreted as pain responses. The tests were performed after at least three habituations at 24 h intervals. The assessments were determined 1 day before surgery for baseline and 1, 3, and 7 days after surgery or injection. All the behavioral tests were performed blinded.

### 2.6. RNA Extraction and Transcriptome Analysis

The total RNA was extracted from mouse spinal cords using TRIzol reagent (Thermo Fisher Scientific, Cat # 15596026, Waltham, MS, USA). For transcriptome analysis, five spinal cords were pooled to synthesize the cDNA library for each group after 24 h of intrathecal injection of gt1b (25 μg/5 μL). The RNA sequencing was conducted by E-Biogen (Seoul, Korea). Briefly, the RNA quality and quantity were evaluated with the Agilent 2100 Bioanalyzer (Agilent, Santa Clara, CA, USA) and NanoDrop 2000 (Thermo Fisher Scientific). The library construction was achieved following the QuantSeq 3′ mRNA-Seq library prep kit FWD (LEXOGEN, Cat # 015.96 Vienna, Austria) manufacturer’s protocol. The cDNA libraries were sequenced on the Illumina NextSeq500 platform (Illumina, San Diego, CA, USA). The gene ontology (GO) enrichment analysis of differentially expressed genes (DEGs) was performed using DAVID 2021 (https://david.ncifcrf.gov/, accessed on 21 December 2021) [17]. The heatmap of RNASeq transcriptome analysis was generated for 23,281 genes after 24 h of i.t. injection of GT1b, and hierarchical clustering analysis (HCL) was conducted using TM4/MeV software (version 4.9.0) to analyze the RNA sequencing data and compare the GT1b-induced transcriptional changes between sexes [18]. The analyzed DAVID GO terms were visualized using a GOcircle plot in MATLAB displaying the fold change of each gene.

### 2.7. Ovariectomy

The eight-week-old female C57BL/6J mice received bilateral ovariectomy (OVX). The mice were anesthetized with isoflurane in an O_2_ carrier (induction at 2% and maintenance at 1.5%) and subsequently subjected to OVX or sham operation via a bilateral back incision. For the OVX group, we excised the anterior uterine horns to remove the ovaries and mitigated bleeding using the High Temp Cautery Kit (FST, Cat # 18010-00, Foster City, CA, USA). The incisions were closed using sterile sutures.

### 2.8. Orchiectomy

The eight-week-old male C57BL/6J mice received bilateral orchiectomy (ORX). The mice were anesthetized with isoflurane in an O_2_ carrier (induction at 2% and maintenance at 1.5%). A midline scrotal incision was performed, and the bilateral spermatic cords were ligated. The testes along with the epididymal adipose were excised from the distal end of the ligature, and bleeding was mitigated using the High Temp Cautery Kit. The incisions were closed using sterile sutures.

### 2.9. ELISA Assay

The mouse blood was collected via a cardiac puncture in microtainer K2E (BD, Cat # 365974, Franklin Lakes, NJ, USA) and centrifuged at 1600× *g* for 15 min at 4 °C to measure the plasma 17β-estradiol levels. The supernatant plasma was collected, and the levels of 17β-estradiol were assessed using the 17 beta Estradiol ELISA kit (Abcam, Cat # ab108667, Cambridge, UK) following the manufacturer’s protocol. To measure the IL-1β release from the GT1b-stimulated primary mixed glia, we treated ATP 30 min prior to supernatants collection to induce the secretion of IL-1β, and collected supernatants of mixed glia centrifuged at 500× *g* for 5 min to discard the cell debris. The IL-1β levels were measured using the mouse IL-1 beta Quantikine ELISA kit (R&D Systems Inc., Cat # MLB00C, Minneapolis, MN, USA) following the manufacturer’s protocol.

### 2.10. In Vitro Primary Mixed Glia Culture

The primary mixed glia cultures were prepared from 1–2-day-old mice, as previously described [19]. In brief, the brain glial cells were cultured in a DMEM, high-glucose formula supplemented with 10% FBS, 10 mM HEPES, 2 mM L-glutamine, 1× penicillin/streptomycin, and 1× nonessential amino acid mixture at 37 °C in a 5% CO_2_ incubator. The media were renewed every 5 days. After 15 days, the primary mixed glia were harvested with 0.25% Trypsin-EDTA and plated in 6-well plates at a density of 5 × 10^5^ cells per well for real-time RT-PCR and ELISA.

### 2.11. Real-Time RT-PCR

The real-time RT-PCR experiments were performed using the StepOnePlus Real-Time PCR system (Applied Biosystems, Foster City, CA, USA) following the 2^−∆∆Ct^ method [20]. The total RNA from the L5 spinal cord tissue and primary mixed glia was extracted using TRIzol (Invitrogen, Carlsbad, CA, USA) and reverse transcribed using TOPscript RT DryMIX (Enzynomics, Cat # RT200, Daejeon, Korea). All the ∆Ct values were normalized to the corresponding GAPDH values and were represented as the fold induction. The following PCR primers were used: *Gapdh* forward, 5′-AGT ATG ACT CCA CTC ACG GCA A-3′; *Gapdh* reverse, 5′-TCT CGC TCC TGG AAG ATG GT-5′; *Il-1β* forward, 5′-GTG CTG TCG GAC CCA TAT GA-3′; *Il-1β* reverse, 5′-TTG TCG TTG CTT GGT TCT CC-3′; *Casp1* forward, 5′-CTG ACA AGA TCC TGA GGG CA-3′; *Casp1* reverse, 5′-AAA GAT TTG GCT TGC CTG GG-3′; *Nlrp3* forward 5′-CCA TCA ATG CTG CTT CGA CA-3′; *Nlrp3* reverse 5′-GAG CTC AGA ACC AAT GCG AG-3′; *Tlr2* forward, 5′- CTC CCA CTT CAG GCT CTT TG -3′; and *Tlr2* reverse, 5′-ACC CAA AAC ACT TCC TGC TG-3′.

### 2.12. Statistical Analysis

The data were analyzed using the Student’s *t*-test for comparisons between two groups. The one- and two-way analysis of variance (ANOVA) with Bonferroni’s post hoc test were used for the statistical analysis of multiple comparisons. All the data are presented as mean ± standard error of the mean (SEM), and differences were considered statistically significant when the *p*-value was < 0.05.

## 3. Results

### 3.1. GT1b-Induced Central Pain Sensitization and Spinal Cord Microglia Activation Are Sexually Dimorphic

To test if GT1b-induced central pain sensitization is sexually dimorphic, we administered GT1b into the spinal cord of male and female mice and compared the mechanical threshold to the von Frey stimuli (Figure 1A). As previously reported, i.t. GT1b administration induced mechanical allodynia 1 and 3 days post-injection (dpi) in the male mice (Figure 1B). However, the female mice were completely resistant to GT1b-induced pain sensitization (Figure 1B), thus indicating sexual dimorphism in GT1b-induced central pain sensitization. We have reported that GT1b activates spinal cord microglia via TLR2, which leads to central pain sensitization [21]. Therefore, we tested TLR2 transcript expression in the spinal cords of female mice, and similar levels of TLR2 transcript were detected in the spinal cords of the female mice compared to the male mice (Figure 1C). To test if there is a gender difference in spinal cord microglia activation upon GT1b administration, we assessed spinal cord microglia activation using Iba-1 immunohistochemistry. The GT1b injection induced spinal cord microglia activation in female mice at 3 dpi, although the activation was not as significant as in male microglia at 1 dpi (Figure 1D,E). We then characterized the morphological features of the GT1b-activated microglia in the female mice by analyzing the soma size, process length, and the branch point of each activated microglia (Figure 1F). In the female mice, GT1b injection increased the microglia soma size (1 dpi) and reduced the process length and branch point number (Figure 1G), which are typical morphological features of the activated microglia [22]. Intriguingly, GT1b injection did not reduce the process length or branch number of spinal cord microglia in male mice; rather, it increased the process length at 1 dpi (Figure 1G). Regarded together, our data indicate that i.t. GT1b administration induces spinal cord microglia activation both in male and female mice, but with distinct kinetics and morphological activation features.

### 3.2. Sexually Dimorphic Transcriptome Profiles of GT1b-Stimulated Mouse Spinal Cords

To further compare the activation features between male and female mouse spinal cord microglia, we analyzed and compared the gene expression profiles in the spinal cord dorsal horn after GT1b administration in the male and female mice, respectively, using RNASeq and hierarchical clustering analysis (Figure 2A). In search of the genes associated with the sexual dimorphism, we screened for differentially expressed genes (DEGs) with expression levels more than two-fold different in the females compared to the males (Figure 2B). To account for differences in the basal gene expression profiles between male and female mice, we conducted additional transcriptome comparisons and annotations using DAVID Gene Ontology (GO) analysis [23,24]. We compared the transcriptome profile between male and female (M-GT1b/M-Veh vs. F-GT1b/F-Veh); however, we did not find any GO terms related to pain (data not shown). Therefore, we compared and annotated the transcriptomes of GT1b-administered males exhibiting pain behavior with that of GT1b-administered females not showing pain behavior. We identified the top ten enriched categories of biological processes (BP), cellular components (CP), and molecular functions (MF) (Figure 2C). Based on DAVID gene ontology analysis, the genes involved in estrogen-related signaling pathways were found to be the most significantly different between the male and female mice (Figure 2D). Meanwhile, the other genes putatively involved in pain sensitization at the spinal cord level (e.g., proinflammatory cytokines, chemokines, etc.) were comparably regulated by GT1b administration (Appendix A). In this regard, we hypothesized that estrogen plays a role in the sexual dimorphism of GT1b-induced central pain sensitization.

### 3.3. GT1b-Induced Central Pain Sensitization Is Dependent on Estrogen

To test whether estrogen plays a role in the GT1b-induced central pain sensitization in female mice, we reduced the systemic estrogen levels in female mice using OVX. Two weeks after removing the ovaries of the 8-week-old adult female mice, uterine shrinkage and an increase in body weight were detected (Figure 3A–C). We also confirmed a decrease in the estrogen levels in the OVX females, and that this decrease was reversed by supplementation with estrogen (OVX−E2) (Figure 3D). While the female mice were resistant to the GT1b-induced central pain sensitization, the OVX rendered female mice susceptible to GT1b-induced mechanical allodynia (Figure 3E). Then, we tested whether exogenous estrogen supplementation could restore the resistance to GT-1b-induced central pain sensitization. When we supplemented these OVX female mice with 17β-estradiol (OVX-E2, 5 μg/kg, 50 μL daily), the OVX mice became resistant again to GT1b-induced mechanical allodynia (Figure 3F). These data indicate that the high estrogen levels in female mice are responsible for the sexual dimorphism of GT1b-induced central pain sensitization. We then tested whether male sex hormones affect GT1b-induced pain sensitization. To this end, we subjected male mice to ORX. Unlike OVX female mice, the ORX male mice exhibited comparable levels of mechanical sensitivity upon GT1b administration (Appendix A), indicating that male sex hormones are not involved in sexual dimorphism.

### 3.4. 17β-Estradiol Inhibits GT1b-Induced Inflammasome Activation and IL-1β Release

Previously, we showed that i.t. GT1b administration induced microglia activation and IL-1β expression in the spinal cord, which results in the development of central sensitization [11]. Since GT1b upregulated IL-1β transcripts not only in male mice but also in female mice (Figure 4A), we tested whether estrogen affects the post-translational modification of IL-1β. IL-1β is initially expressed in its pro-form (pro-IL-1β) and then released upon cleavage by NLRP3- and Caspase-1-containing inflammasomes [21]. Therefore, we compared the gene expression of NLRP3 and Caspase-1 in male and female mice upon GT1b stimulation. Though GT1b administration induced NLRP3 expression in both sexes, Caspase-1 was upregulated only in male mice (Figure 4B,C), indicating sexually dimorphic inflammasome activation. Then, we examined the estrogen effects on primary mixed glia in vitro (Figure 4D). We treated cells with GT1b (10 μg/mL) for 16 h, following pre-treatment with 17β-estradiol (100 nM) for either 16 h or 0.5 h. To measure the IL-1β secreted from mixed glia, we pretreated all groups, including the control, with ATP (5 mM) 30 min prior to collecting the supernatant. While 17β-estradiol pretreatment did not affect GT1b-stimulated Caspase-1 and IL-1β transcript expression, it significantly inhibited the IL-1β release from the primary mixed glia (Figure 4E–G). Regarded together, these results suggest that estrogen inhibits IL-1β release in the spinal cords of female mice, which underlies the sexual dimorphism of GT1b-induced central pain sensitization.

## 4. Discussion

In this study, we discovered that GT1b, a previously identified endogenous TLR2 agonist used in nerve-injury-induced neuropathic pain, induces sexually dimorphic central pain sensitization; it induced pain sensitization only in male but not in female mice. Sexual dimorphism in central pain sensitization has been reported in several other animal models. In LPS-induced pain, i.t. administration of LPS induced neuropathic pain only in male mice [7], while local LPS administration in the paw induced pain in both male and female mice [25]. Likewise, i.t. CSF1 administration induced pain sensitization only in male mice [8,9]. Our data are in line with these previous studies. Considering TLR2 and TLR4 share most of their downstream intracellular signaling pathways mediated by MyD88 [26], it is not surprising that the GT1b-mediated activation of microglial TLR2 induces central sensitization only in male mice, not female mice. Our data, along with previous reports, indicate that microglia activation in female mice is not sufficient to induce central pain sensitization [27]. Similarly, nerve injury-induced spinal cord microglia activation is critical for the development of neuropathic pain only in male mice. However, it is not required for nerve injury-induced neuropathic pain in female mice [28]. Instead, in female mice, central pain sensitization is mediated by T cells recruited to the spinal cord after nerve injury [6]. In this regard, it is conceivable that i.t. injection of GT1b activates spinal cord microglia but does not recruit T cells into the spinal cord, and, thus, it fails to induce central pain sensitization in female mice.

In the search for mechanisms underlying sexually dimorphic functions of male spinal cord microglia vs. female microglia activation, we investigated sex-specific transcriptome profiles in the spinal cord upon GT1b injection. Previous transcriptome analyses to identify DEGs in microglia responsible for sexual dimorphism have provided minimal insight into the mechanisms [10]. According to this study, upon chronic constriction sciatic nerve injury, most genes involved in microglia activation and implicated in central pain sensitization, such as proinflammatory cytokines, were also upregulated in female microglia [10]. Though it failed to identify specific genes that were selectively upregulated in male microglia and rendered central pain sensitization, male microglia displayed more prominent global transcriptional shifts and increased phagocytic activity compared to female microglia [10]. Likewise, our RNAseq results revealed that GT1b regulates genes involved in neuropathic pain in both sexes. Meanwhile, our DAVID gene ontology analysis indicated that most enriched DEGs in biological processes are associated with the response to estrogen, including *F7*, *Tph2*, *Cyp27b1*, *Krt19*, *Abcc2*, *Mstn*, *Dhh*, *Ghrl*, *Smad6*, *Agtr1a*, *Tshb*, and *Agtr1b*, which suggests the putative involvement of estrogen in the sexual dimorphism of GT1b-induced central pain sensitization. Therefore, we focused on the estrogen response, and our study using OVX mice demonstrated that estrogen was indeed responsible for the lack of pain-sensitizing effects of i.t. GT1b administration in female mice.

Estrogen is well-known for its anti-inflammatory and neuroprotective effects on the nervous system. In stroke and ischemic brain injury, estrogen exerts neuroprotective functions by inhibiting inflammasome activation and proinflammatory cytokine expression in the brain [29,30]. In addition, estrogen attenuates the spinal-cord-injury-induced inflammatory response by regulating inflammasomes [31]. Studies indicate that estrogen mediates its effects by affecting glia activation. For instance, estrogen inhibits global cerebral-ischemia-induced NLRP3 inflammasome activation and proinflammatory cytokine expression in glia [32]. It also inhibits spinal-cord-injury-induced microglial p38 and ERK activation, astrocyte JNK activation, and thereby mediates the anti-inflammatory effects in the spinal cord [33]. In line with these previous studies, our data revealed that estrogen inhibits GT1b-induced IL-1β production in primary mixed glia. It was well-known that IL-1β expression in the spinal cord renders central pain sensitization [34,35]. Of note, IL-1β transcripts in the spinal cords of female mice were comparable to those of male mice. Instead, the IL-1β released into the media was inhibited by estrogen, and the Caspase-1 induction by GT1b was blocked in the female mice. It must be noted that GT1b-stimulated caspase-1 expression was not inhibited by E2. We speculate that E2 may indirectly suppress glial caspase-1 transcript in vivo. Although we revealed the essential role of E2 in the sexual dimorphism of the GT1b-induced central sensitization, the functional and mechanistic link between the results obtained in primary glial cultures and in vivo remains weak, which is a limitation of this study. Therefore, further studies are needed on the regulation of GT1b-induced inflammasome activation by E2 to completely elucidate the mechanism of sexual dimorphism in the GT1b-induced central sensitization (Figure 5).

While estrogen exerts anti-inflammatory and neuroprotective effects on the nervous system, it has also been shown to enhance pain via the modulation of gene expression [36] and neuronal activation in dorsal root ganglia (DRG) of females [37]. Therefore, further studies are needed to fully elucidate the mechanisms underlying the biphasic effects of estrogen on pain modulation in different contexts. Anyhow, our study used an i.t. GT1b-induced pain model, which primarily involves upper circuits and may not directly affect DRGs.

The sexual dimorphic mechanism of GT1b-induced central pain sensitization that we suggest is distinct from that of TLR4-induced pain. According to a study by Sorge et al., TLR4-mediated pain is dependent on testosterone [7]. However, in our study, GT1b induced comparable pain in ORX mice, and, therefore, GT1b-induced pain sensitization was independent of testosterone. Thus far, it is not clear why there is such a difference. Although TLR2 and TLR4 share MyD88 as their intracellular signaling molecule, TLR4 induces an additional intracellular signal via IRF3 [26]. Therefore, it is speculated that IRF3-dependent microglia activation is affected by testosterone, which needs to be tested in the future.

Recent studies indicate that morphological phenotypes of microglia, such as ramified, rod-like, activated, and amoeboid forms, represent the state of microglia and can be used as indicator of the CNS physiological environment [22,38]. Here, we report that GT1b-induced microglial activation occurs in both males and females, but with different kinetics and morphology. These features might be involved in the E2-mediated suppression of inflammasome activation and IL-1β secretion.

In summary, our study revealed that i.t. GT1b administration induces central pain sensitization in a sexually dimorphic manner. We discovered that estrogen is responsible for sexual dimorphism, and that estrogen ameliorates GT1b-induced IL-1β production in the spinal cord by inhibiting inflammasome activation as an underlying mechanism. Our study may elucidate sex-specific therapeutic strategies to resolve central pain sensitization utilizing estrogen.

## Figures and Tables

**Figure 1 cells-12-00808-f001:**
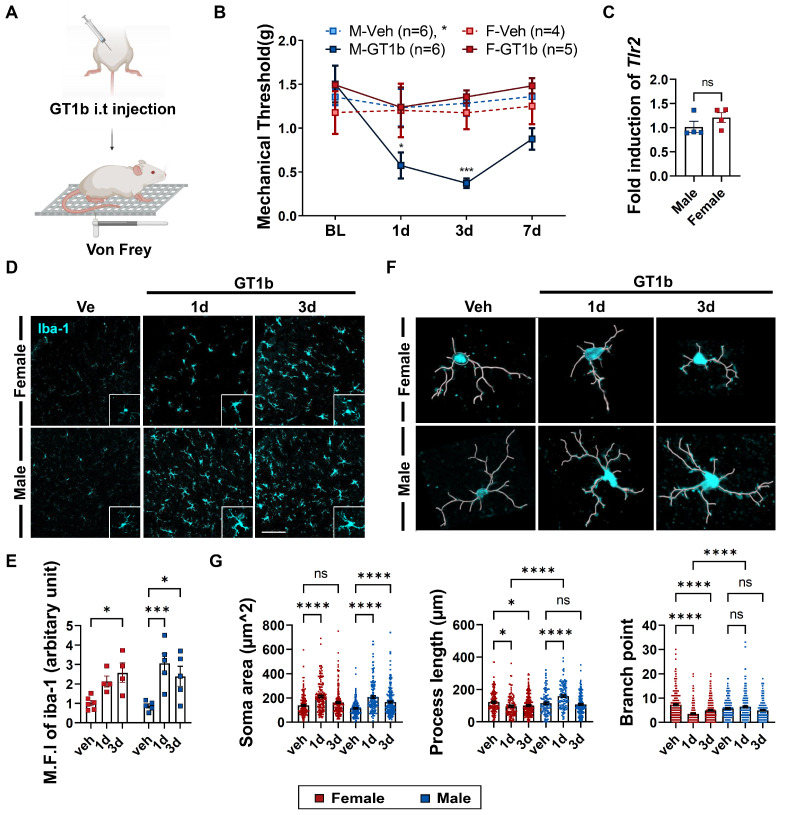
Characterization of GT1b-induced central pain sensitization and GT1b-induced microglial activation. (**A**) Schematic diagram of i.t administration of GT1b and von Frey allodynia test. (**B**) Mechanical threshold of von Frey tests on hind paws after GT1b administration (n = 4 to 6/group, two-way ANOVA with Tukey’s multiple comparison test post hoc, * vs. M-Veh). (**C**) Relative *TLR2* transcript levels in the spinal cords of female mice compared to male mice. (**D**) Representative image of a spinal cord immunostained with Iba-1 (Scale bar, 100 μm) and (**E**) mean fluorescence intensity of Iba-1 (n = 4 to 6/group, two-way ANOVA with Bonferroni’s multiple comparison test post hoc). (**F**) Representative image of microglial morphology analyzed using IMARIS. (**G**) Soma area, process, and branch points of microglia (n = from 139 to 235 microglia/group, two-way ANOVA with post hoc Tukey’s multiple comparison test). Data are presented as mean ± SEM. ns: not significant, * *P* < 0.05, *** *P* < 0.001, and **** *P* < 0.0001. BL: basal lever. 1, 3, 7d: 1, 3, 7 days after injection. Veh: vehicle control group. M.F.I: mean fluorescence intensity.

**Figure 2 cells-12-00808-f002:**
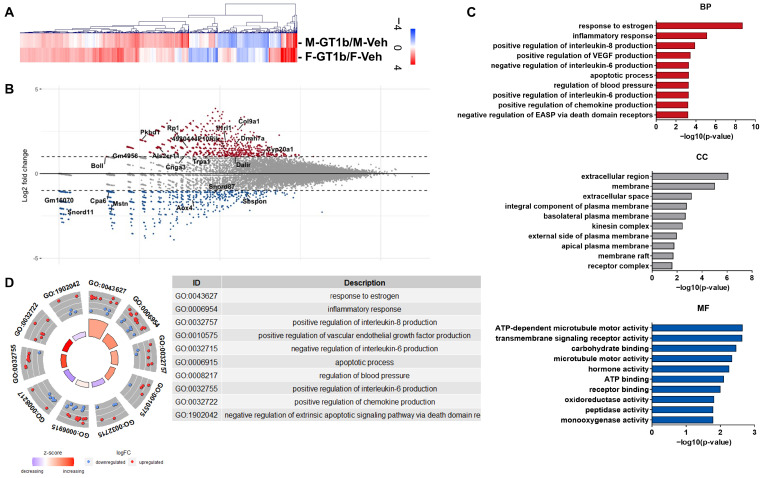
Transcriptomic analysis of GT1b-injected mouse spinal cords. (**A**) Heat map of RNASeq transcriptome analysis for 23281 genes after 24h of i.t. injection of GT1b (**B**) MA plot for DEGs of GT1b-stimulated female compared to GT1b-stimulated male mice. Greater than two-fold upregulated or downregulated genes are plotted. Y-axis shows the fold induction of DEGs, and the X-axis shows the basal expression level of each gene. (**C**) The top ten enriched GOs were analyzed using DAVID functional gene ontology analysis of DEGs (BP: Biological process, CC: cellular component, and MF: molecular function). (**D**) GOcircle plot displaying the fold change (logFC) of each gene in the top ten enriched BP GO terms. The chart displays the annotation categories of each GO. Z-scores are displayed in the inner circles. M-GT1b: GT1b administered male. M-Veh: vehicle administered male. F-GT1b: GT1b administered female. F-Veh: vehicle administered female.

**Figure 3 cells-12-00808-f003:**
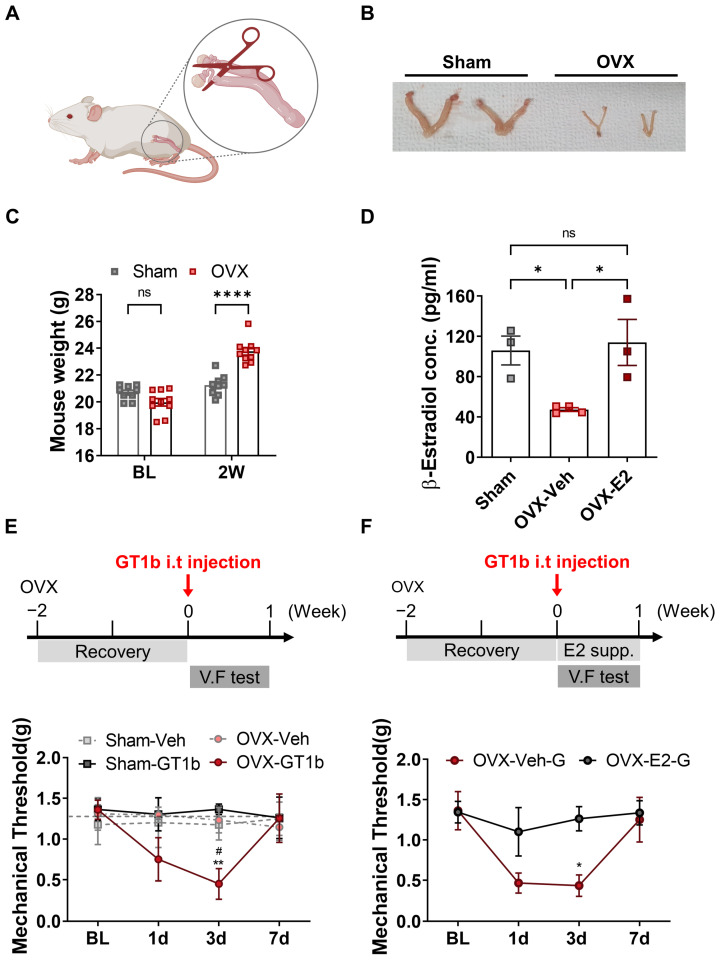
GT1b-induced mechanical allodynia relapse after OVX in female mice, which was reversed by estrogen supplement. (**A**) Schematic diagram of bilateral ovariectomy. (**B**) Representative images of uteri in sham and OVX mice 2 weeks after surgery. (**C**) Mouse weight (n = from 9 to 10/group, two-way ANOVA with post hoc Bonferroni’s multiple comparison test), and (**D**) serum levels of estradiol were measured 2 weeks after OVX (n = from 3 to 4/group, one-way ANOVA with post hoc Tukey’s multiple comparison test). (**E**) Mechanical allodynia was measured after GT1b i.t injection in animals with OVX (n = from 4 to 5/group, two-way ANOVA with post hoc Tukey’s multiple comparison test, # vs. Sham-GT1b, * vs. OVX-Veh), and (**F**) OVX—E2 supplementation (n = from 3 to 5/group, two-way ANOVA with post hoc Bonferroni’s multiple comparison test, # vs. OVX-Veh-V, * vs. OVX-E2-G). Data are presented as mean ± SEM. ns: not significant, * *P* < 0.05, ** *P* < 0.01, and **** *P* < 0.0001. OVX: ovaryectomy. Sham: control for ovarectomy. OVX-Veh-G: GT1b administered vehicle control group of ovaryectomized female. OVX-E2_G: GT1b administered E2 supplement group of ovaryectomized female. BL: basal level. 2W: 2 weeks.

**Figure 4 cells-12-00808-f004:**
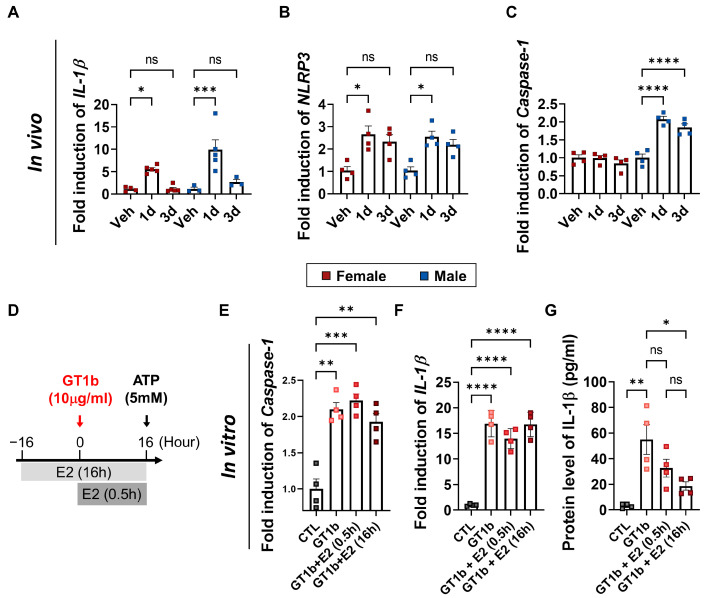
17β-estradiol regulates post-translational IL-1β modification. Transcript levels of (**A**) IL-1β, (**B**) NLRP3, and (**C**) Caspase-1 in the spinal cord after GT1b i.t. injection (n = from 3 to 5/group, two-way ANOVA with post hoc Tukey’s multiple comparison test). (**D**) Experimental scheme of the IL-1β release assay from primary glia. (**E)** Transcript levels of Caspase-1, (**F**) IL-1β, and (**G**) protein levels of released IL-1β from primary glia (n = 4/group, one-way ANOVA with post hoc Tukey’s multiple comparison test). Data are presented as mean ± SEM. ns: not significant, * *P* < 0.05, ** *P* < 0.01, *** *P* < 0.001, and **** *P* < 0.0001.

**Figure 5 cells-12-00808-f005:**
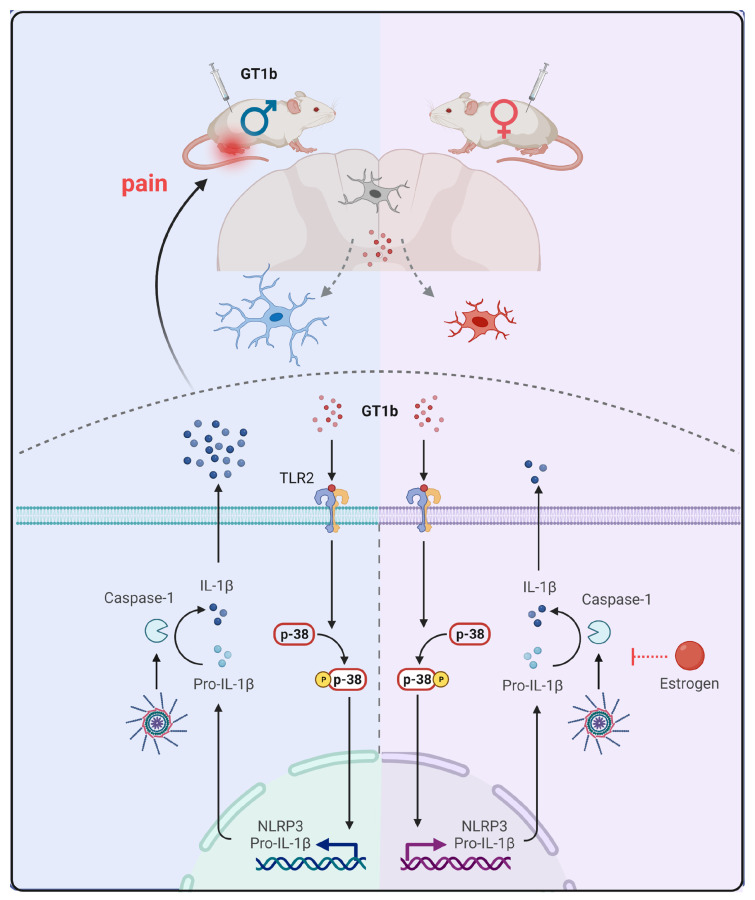
Sexual dimorphic mechanism of GT1b-induced pain central sensitization.

## Data Availability

The datasets used and/or analyzed during the current study are available from the corresponding author on reasonable request.

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
