# Peer review of "Estrogen Mediates the Sexual Dimorphism of GT1b-Induced Central Pain Sensitization"

_cells, 2023, doi:10.3390/cells12050808_

Round 1
Reviewer 1 Report
Previous stud from this group reported that GT1b, a ganglioside, acts as an endogenous agonist of TLR2 and is sufficient to induce microglia activation and pain sensitization. In this study, the authors investigated sex dimorphism of GT1b induced microglia activation and mechanical pain. They found that intrathecal GT1b administration induced mechanical pain in male not female mice. Interestingly, they showed that estrogen is responsible for masking pain following GT1b injection. However, male hormones are not involved. Mechanistically, estrogen may inhibit GT1b-induced inflammasome activation and IL-1b production. The findings are very interesting and provide new insights into sex dimorphism in pain.
Minor comments:
Please enlarge Fig. 2B to show all the details.
I wonder if GT1b can induce heat hyperalgesia.
Estrogen has also been shown to enhance pain via modulation of gene expression and neuronal activation in DRGs of females. Some discussion is helpful.
Line 332: It is very interesting to show that most enriched DEGs in biological processes are associated with the response to estrogen including F7, Tph2, Cyp27b1, Krt19, Abcc2, Mstn, Dhh, Ghrl, Smad6, Agtr1a, Tshb, and Agtr1b. Are these genes known to suppress pain?
Author Response
Thank you very much for the opportunity to revise our manuscript. We tried our best to respond to the reviewer's queries and comments.

Reviewer 2 Report
Jaesung Lee and colleagues in a previous work identified that GT1b induces mechanical hypersensitivity in males but not in females, while GT1b induces microglial proliferation in both sexes (Jaesung Lee et al, Molecular Pain, 2021). In this work, they aimed to investigate the mechanism underlying the sexual dimorphic effect of GT1b on central pain sensitization. In my opinion, important information regarding how the experiments were executed and an appropriate description of the results are missing, thus the article does not provide sufficient information to critically analyze the results and evaluate the quality of the work.
What doses of GT1b were used in the animal experiments?
"Results" and "methods" sections related to the transcriptome analysis do not give sufficient information to analyze critically this part, these parts should be drastically revised. Here are some examples: How many samples were analyzed? The samples were taken on which day after ganglioside administration? how was made the comparison? Given the description in the article, it is really difficult to understand, seem that DEGs were identified by analyzing treated female vs treated male; if you compare females against males it is expected to identify an increase in estrogen-related pathways; on the contrary, whether a more consistent analysis was performed (eg Vehicle F/GT1b F vs Vehicle M/GT1b M) the type of comparison performed must be clarified. The information about how graphs 2A and 2B were generated and their description are not sufficient. Graph 2B is too small to be read. Figure 2D and which kind of software was used to obtain should be explained……
To explain the absence of pain sensitization in females they "postulate that estrogen in female mice inhibits GT1b-induced inflammasome activation and the subsequent release of IL-1b in the spinal cord, which renders female mice refractory to GT1b-induced central pain sensitization”. In my opinion, the functional link between the results obtained in primary glial cultures and in vivo is weak and should be strengthened with additional supportive experiments. The experiment performed with primary mixed glia is not well described, and important information such as the dose of E2 and GTB1 and for how long the cells were treated are missing. Are the doses of E2 and GTB1 used representative of the in vivo physiology/experiments? In primary glia is caspase-1 expression upon GTB1 administration affected by E2? The inhibition of caspase-1 activity (or expression) in vivo is sufficient to make males resistant to GT1b-induced mechanical allodynia?
What means “ATP” in figure 4D?
Microglia activation upon GT1b administration in females and males was already reported in their previous work (Jaesung Lee et al 2021); in this work, the kinetic of activation and the changes in microglial morphology were characterized in more detail but it is not clear what advancement of the current knowledge bring this analysis.
The Materials and Methods section needs to be implemented with sufficient information to reproduce the experiments. Particularly critical are the “Real-time RT-PCR” and "transcriptome" paragraphs. The catalogue number of important reagents should be inserted to understand the type of reagent used (eg antibody).
It is not clear why it is important to study the sexual dimorphic effect of GT1b.
GT1b should be more extensively described in the introduction.
Female mice are not mentioned in “2.1 Animals”.
Figure 3D, column OVX-E2 should be better described in the text.
Some abbreviations inserted into the graphs should be described in the figure legend (eg BL, 2W).
Author Response

(The authors gave the same response as above.)

Round 2
Reviewer 2 Report
I could not find in the revised text the dose of 17beta-estradiol, GT1b, and ATP used in the cellular experiments. I suggest inserting it into chapter 3. 4. Also specify for how long the cells were treated with each molecule. Specify if CTL cells were treated with ATP. If the CTL cells were not treated with ATP, describe why in the text and highlight this limitation.
In my opinion, the functional/mechanistic link between the results obtained in primary glial cultures and in vivo remains weak (chapter 3.4). Since the authors have the data relative to the caspase-1 expression in primary glia " In primary glia, caspase-1 expression upon GT1b stimulation was not affected by E2 treatment" insert this data into the article and explain the hypothesis regarding the divergence between the cellular and animal data results. Since the Authors did not perform experiments to better support the mechanism hypotheses, I suggest inserting the limitations of the experiments performed; and underlining that other studies are needed to clarify the interconnection between inflammasome/E2/Gt1b.
"We compared four groups in the transcriptome analysis: GT1b administered male, Saline administered male, GT1b administered female and saline administered female. We have originally compared the transcriptome profile between male and female as suggested (e.g. Vehicle F/GT1b F vs Vehicle M/GT1b M), but we could not find any gene ontology (GO) terms related to pain in that analysis (data not shown). Therefore, then, we compared the transcriptome of GT1badministered males showing pain behaviour vs. GT1b-administered females showing no pain behaviour. Considering male and female have different basal level gene expression profile, although we did not find GO terms in between Vehicle F/GT1b F vs Vehicle M/GT1b M, we assumed that GO term between GT1b-male vs. GT1b-female gives information regarding on pain sensitivity." these concepts have to be inserted into the article to allow the reader to understand the type of analysis performed.
Fig 2D has low resolution
"Here, we report that GT1binduced microglial activation occurs in both males and females, but with different kinetics and morphology. These features might be involved in the E2-mediated suppression of inflammasome activation and IL-1 secretion." I think that this concept could be inserted into the discussion.
Specify the acronym DRG.
Author Response
Again, we thank the reviewer for their comments and suggestions, and we look forward to hearing back from you at your earliest convenience.
